# Self-supervised learning for label-free segmentation in cardiac ultrasound

Danielle L. Ferreira [1,2], Connor Lau[1,2], Zaynaf Salaymang[1] & Rima Arnaout [1,2,3,4]

Segmentation and measurement of cardiac chambers from ultrasound is critical, but laborious and poorly reproducible. Neural networks can assist, but supervised approaches require the same problematic manual annotations. We build a pipeline for self-supervised segmentation combining computer vision, clinical knowledge, and deep learning. We train on 450 echocardiograms and test on 18,423 echocardiograms (including external data), using the resulting segmentations to calculate measurements. Coefficient of determination ($r^2$) between clinically measured and pipeline-predicted measurements (0.55-0.84) are comparable to inter-clinician variation and to supervised learning. Average accuracy for detecting abnormal chambers is 0.85 (0.71-0.97). A subset of test echocardiograms ($n = 553$) have corresponding cardiac MRIs (the gold standard). Correlation between pipeline and MRI measurements is similar to that of clinical echocardiogram. Finally, the pipeline segments the left ventricle with an average Dice score of 0.89 (95% CI [0.89]). Our results demonstrate a manual-label free, clinically valid, and scalable method for segmentation from ultrasound.

Artificial intelligence (AI) has the potential to revolutionize medical imaging. The revolution will not be supervised[1].

Nowhere is the potential for AI, as well as the burden of supervised learning, more clear than for cardiac ultrasound, the primary cardiac imaging modality[2]. The quantification of chamber size, mass, and function are critical to diagnosis, prognosis, and management[3]. However, this quantification is laborious, requiring several manual annotations per exam. Furthermore, even when performed by experts, manual annotations can be susceptible to inter- and intra-observer variability given the low spatial resolution and artifacts inherent to ultrasound imaging[4]. Measurement variability can compound e.g. when linear and area measurements are used to calculate volumes and function[5,6]. Finally, despite the importance of structural and functional measurements for all chambers[3], the right heart and left atrium are often neglected in practice, in large part due to the laborious nature of performing additional annotations.

To overcome these challenges, researchers have turned to deep learning-based semantic segmentation. To date, supervised approaches have been used for this task[7–11], but these require the same manual annotations mentioned above. Therefore, supervised segmentation does not alleviate labeling burden, and instead raises additional concern of using variable and error-prone manual annotations as ground-truth[12]. In fact, to mitigate potential bias from any one person, multiple labelers are advocated, which further increases the cost of labeling[13]; an entire industry has arisen to perform manual labeling[14]. Manual labeling also scales poorly with each additional structure to be labeled, perhaps explaining why most studies of semantic segmentation of the heart to date have focused on the left ventricle[9,15–18]. Emerging foundation models for segmentation of photographic images do not obviate manual labeling (some requiring over a billion manual annotations)[19], require additional manual input at the point of use, and do not work well on ultrasound without additional labor[20].

[1]Department of Medicine, Division of Cardiology, University of California, San Francisco, 521 Parnassus Avenue, San Francisco, CA, USA. [2]Bakar Computational Health Sciences Institute, University of California, San Francisco, 490 Illinois St, San Francisco, CA, USA. [3]Department of Radiology, Center for Intelligent Imaging, 505 Parnassus Avenue, San Francisco, CA, USA. [4]UCSF-UC Berkeley Joint Program in Computational Precision Health, 505 Parnassus Avenue, San Francisco, CA, USA. e-mail: rima.arnaout@ucsf.edu

Self-supervised learning (SSL) has the potential to obviate these problems. Broadly defined, SSL learns to perform a task that conventionally requires supervised learning, but instead uses information generated from the data itself rather than relying on manual human labels. SSL networks are trained with automatically generated labels and human annotation is not required[1,21–24]. SSL has been used to segment objects in photographic imaging with some success[25], but has been rare in biomedical imaging to date due to the low acceptable margin of error required for such applications[26,27] (Supplementary Fig. S2). In noisy ultrasound, automated segmentation has proved to be especially challenging[28–30], with some approaches to date reducing, but not obviating, use of manual labels[10,11].

To bridge the gap between supervised segmentation and the challenges of ultrasound imaging, we show that weak labels can be created using computer vision techniques, circumventing both the time-consuming and subjective nature of human labels. We further show that using these labels in a self-supervised deep learning pipeline designed to mitigate overfitting and incorporate clinical domain knowledge can segment echocardiograms with performance rivaling clinicians.

## Results
### Overview
We developed a pipeline (Fig. 1) to provide self-supervised segmentation of echocardiograms. We first extracted weak labels for cardiac chambers using traditional computer vision techniques and clinical shape priors (Methods, Supplemental Methods). We then used these weak labels to train more accurate segmentations, utilizing early stopping and clinical domain-guided label refinement in successive training steps to achieve final performance. Segmentation predictions from the final step were used to calculate biometrics for each chamber[3], focusing on the most clinically relevant[31] views: the apical 2-chamber (A2C), apical 4-chamber (A4C), and short-axis mid (SAX).

We used 93,000 images from 450 transthoracic echocardiograms for training and validation and 4,476,266 images from 8393 echocardiograms for testing, across a range of image qualities, pathologies, and patient characteristics (Table 1, "all-comers"). A subset of all-comers ($n = 553$) had corresponding cardiac MRI (CMR) available within 30 days for additional comparison among to CMR, as CMR is considered the gold standard for cardiac measurements (Table 1, "CMR subset"). We also tested our pipeline against an external dataset of A4C images with an additional 10,030 patients.

### Self-supervised pipeline can learn to segment ultrasound
After confirming reports[29,32] that chamber measurements derived from computer vision alone correlate poorly with clinical measures (Supplementary Fig. S2, initial steps in Figs. 2–5), we developed the supervised learning pipeline (Fig. 1) described above and in the Methods. Figures 2–5 show examples of segmentation performance through successive steps in the pipeline for all three views, showing that subsequent steps in the pipeline can correct initial errors and learn rather than fail, even on images across a range of qualities, pathologies, and derangements.

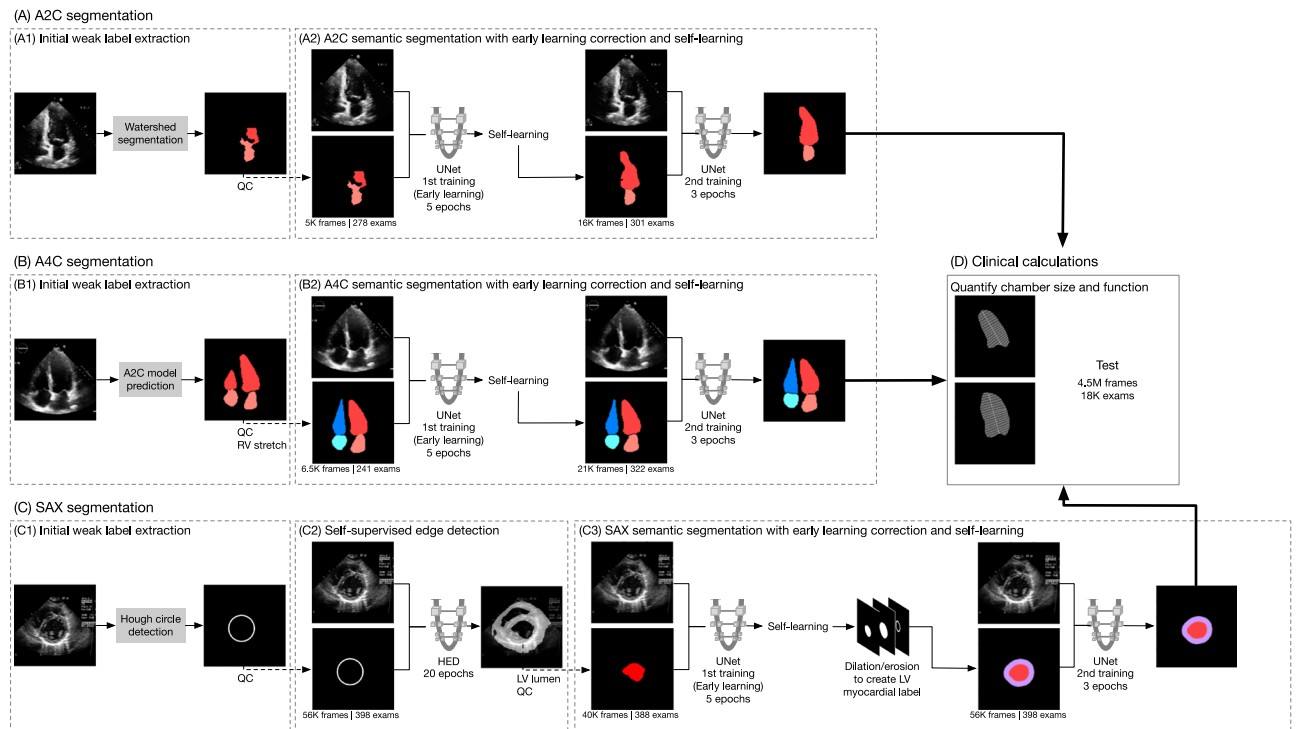

**Fig. 1 | Overview of the self-supervised segmentation pipeline. A** A2C segmentation. **A**1 Initial weak label creation using computer vision. **A**2 Semantic segmentation training includes a first training with early learning and self-learning to arrive at a second and final trained model used to run inference on test data. **B** A4C segmentation. **B**1 Initial weak label creation from inference using the A2C final model from (**A**), resulting in predictions of two "LA" and two "LV" chambers per image. **B**2 Predictions from B1 were reassigned four chamber labels, and clinically-guided morphological operations were performed to stretch the RV apex to known correlations of RV and LV length. Early learning and self-learning were then performed to arrive at the final A4C model. **C** SAX segmentation. **C**1 Initial weak labels were generated through Hough circle detection and **C**2 a holistic edge detection (HED) network was trained (early learning). **C**3 The prediction from (**C**2) was then filled in and used as a label to train a first UNet. After self-learning to recruit more usable labels, dilation and erosion were applied to the UNet prediction to create labels for the epicardial and endocardial areas. A second UNet was trained with these labels, resulting in the final model. **D** Clinical calculations. Predictions from (**A**2), (**B**2), and (**C**3) were used to compute chamber dimensions, areas, volumes, and Dice scores on all-comers and external test datasets, respectively. Number of images, exams, and training epochs specified for each step of the pipeline. LV left ventricle (red), LA left atrium (light blue), RV right ventricle (dark blue), RA right atrium (pink), HED holistically nested edge detection, A2C apical 2-chamber, A4C apical 4-chamber, SAX short-axis mid.

**Table 1 | Study population demographics**

| | Training & validation set (n = 450) | Holdout test set (n = 8393) | | All-comers vs. Training p-value (mwu) | All-comers vs. CMR subset p-value (mwu) | Training vs. all-comers vs. CMR subset p-value (kruskal) | External test set (n = 10,030) |
|---|---|---|---|---|---|---|---|
| | | All-comers (n = 8393) | Subset with CMR (n = 553) | | | | |
| **Demographics** | | | | | | | |
| Age, years ± s.d.(range) | 58 ± 17 (20–90) | 60 ± 17 (15–120) | 49 ± 16 (19–88) | <0.001 | <0.001 | <0.001 | 68 ± 21 |
| Female, n (%) | 220 (49%) | 3552 (50%) | 193 (40%) | 0.5 | <0.001 | <0.001 | 4885 (48%) |
| White, n (%) | 238 (55%) | 3590 (52%) | 246 (54%) | 0.4 | 0.8 | 0.7 | - |
| Asian, n (%) | 69 (16%) | 1358 (20%) | 63 (14%) | 0.04 | 0.001 | <0.001 | - |
| Latinx, n (%) | 56 (13%) | 850 (12%) | 74 (16%) | 0.8 | 0.03 | 0.09 | - |
| Black or African American, n (%) | 31 (7%) | 567 (8%) | 35 (8%) | 0.4 | 0.6 | 0.6 | - |
| Other, n (%) | 39 (9%) | 541 (8%) | 40 (9%) | 0.5 | 0.6 | 0.7 | - |
| **Machine manufacturer** | | | | | | | |
| Philips | 335 (74%) | 4826 (69%) | 295 (62%) | 0.009 | 0.002 | <0.001 | 10003 (100%) |
| GE | 60 (13%) | 2029 (29%) | 27 (6%) | <0.001 | <0.001 | <0.001 | - |
| Siemens | 55 (12%) | 182 (3%) | 155 (32%) | <0.001 | <0.001 | <0.001 | - |
| **Clinicians** | | | | | | | |
| Number of unique sonographers, n | 26 | 53 | 30 | - | - | - | - |
| Number of unique diagnosing physicians, n | 17 | 34 | 23 | - | - | - | - |
| **Study quality** | | | | | | | |
| Fair, poor, or technically difficult (%) | 123 (27%) | 2438 (35%) | 64 (13%) | 0.002 | <0.001 | <0.001 | - |
| **Cardiac disease** | | | | | | | |
| Arrhythmia, n (%) | 127 (28%) | 2062 (29%) | 126 (26%) | 0.6 | 0.2 | 0.4 | - |
| Heart failure including heart transplant, n (%) | 126 (28%) | 1882 (27%) | 184 (39%) | 0.6 | <0.001 | <0.001 | 2874 (29%) |
| Dilated cardiomyopathy, n (%) | 4 (1%) | 110 (2%) | 14 (3%) | 0.3 | 0.02 | 0.03 | - |
| Restrictive cardiomyopathy, n (%) | 1 (0%) | - | 3 (1%) | 0.6 | 0.001 | 0.03 | - |
| CAD risk factors & CAD equivalents[a], n (%) | 261 (58%) | 4128 (59%) | 228 (48%) | 0.8 | <0.001 | <0.001 | - |
| CAD, MI, cardiac arrest, n (%) | 86 (19%) | 1495 (21%) | 89 (19%) | 0.3 | 0.2 | 0.2 | 2290 (23%) |
| Significant valve disease[b], n (%) | 63 (14%) | 1041 (15%) | 62 (13%) | 0.6 | 0.3 | 0.5 | - |
| Significant aortic stenosis[b], n (%) | 15 (3%) | 219 (3%) | 11 (2%) | 0.8 | 0.3 | 0.6 | - |
| Significant aortic regurgitation[b], n (%) | 9 (2%) | 80 (1%) | 18 (4%) | 0.1 | <0.001 | <0.001 | - |
| Pericardial disease, n (%) | 27 (6%) | 209 (3%) | 34 (7%) | <0.001 | <0.001 | <0.001 | - |
| Significant pericardial effusion[b], n (%) | 7 (2%) | 69 (1%) | 20 (5%) | 0.2 | <0.001 | <0.001 | - |
| Aortic disease, n (%) | 5 (1%) | 138 (2%) | 13 (3%) | 0.2 | 0.2 | 0.2 | - |
| Pulmonary HTN, n (%) | 42 (9%) | 531 (8%) | 45 (9%) | 0.2 | 0.1 | 0.1 | - |
| Congenital Heart Disease, n (%) | 28 (6%) | 327 (5%) | 68 (14%) | 0.1 | <0.001 | <0.001 | - |
| Cardiac Prostheses[c], n (%) | 59 (13%) | 1011 (14%) | 78 (16%) | 0.5 | 0.2 | 0.4 | - |
| **Measurements** | | | | | | | |
| BSA, m² mean ± s.d.(range) | 1.88 ± 0.28 (1.1–2.9) | 1.85 ± 0.26 (0.7–4.3) | 1.89 ± 0.25 (1.2–2.6) | 0.09 | <0.001 | <0.001 | - |
| LV ejection fraction, %, mean ± s.d.(range) | 61 ± 12 (8–83) | 59 ± 12 (4–89) | 55 ± 16 (10–83) | 0.02 | <0.001 | <0.001 | 56 ± 13 |

**Table 1 (continued) | Study population demographics**

| | Training & validation set (n = 450) | Holdout test set (n = 8393) | | All-comers vs. Training p-value (mwu) | All-comers vs. CMR subset p-value (mwu) | Training vs. all-comers vs. CMR subset p-value (kruskal) | External test set (n = 10,030) |
|---|---|---|---|---|---|---|---|
| | | All-comers (n = 8393) | Subset with CMR (n = 553) | | | | |
| LV ejection fraction <35%, n (%) | 29 (7%) | 444 (6%) | 75 (16%) | 0.9 | <0.001 | <0.001 | 948 (8%) |
| LV ejection fraction abnormal (qualitative), n (%) | 79 (18%) | 1454 (21%) | 170 (36%) | 0.1 | <0.001 | <0.001 | – |
| LV diastology abnormal, n (%) | 281 (65%) | 2551 (38%) | 230 (56%) | 0.2 | 0.01 | 0.02 | – |
| LV end diastolic volume index, mL/m², mean ± s.d.(range) | 56 ± 23 (19–211) | 54 ± 24 (10–244) | 64 ± 29 (10–234) | 0.09 | <0.001 | <0.001 | 91 ± 46[d] |
| LV end systolic volume index, mL/m², mean ± s.d.(range) | 24 ± 19 (5–172) | 24 ± 19 (3–203) | 32 ± 25 (3–189) | 0.6 | <0.001 | <0.001 | 43 ± 35[d] |
| LV Size abnormal, n (%) | 56 (13%) | 929 (14%) | 126 (28%) | 0.6 | <0.001 | <0.001 | – |
| LV mass index, g/m², mean ± s.d.(range) | 91 ± 34 (34–290) | 86 ± 26 (19–238) | 98 ± 30 (48–195) | 0.04 | 0.002 | 0.001 | – |
| LV mass abnormal, n (%) | 106 (29%) | 1565 (26%) | 127 (34%) | 0.3 | 0.001 | 0.003 | – |
| LA volume index, mL/m², mean ± s.d.(range) | 31 ± 14 (10–98) | 31 ± 13 (8–142) | – | 0.3 | <0.001 | – | – |
| RA volume, mL, mean ± s.d.(range) | – | 22 ± 13 (4–143) | – | – | – | – | – |
| RV end diastolic area, cm², mean ± s.d.(range) | – | 17 ± 5 (7–40) | – | – | – | – | – |
| RV end systolic area, cm², mean ± s.d.(range) | – | 9 ± 4 (3–28) | – | – | – | – | – |

LV left ventricle, LA left atrium, RV right ventricle, RA right atrium, BSA body surface area, mwu Mann–Whitney U test, kruskal Kruskal–Wallis test.
aCAD risk factors include hypertension, hyperlipidemia, smoking, family history of CAD, drug use. CAD equivalents include diabetes, peripheral vascular disease, and cerebrovascular disease.
bSignificant disease includes any severity greater than mild.
cIncludes pacemakers, grafts, balloon pumps, ventricular assist devices.
dValues are not indexed by BSA.

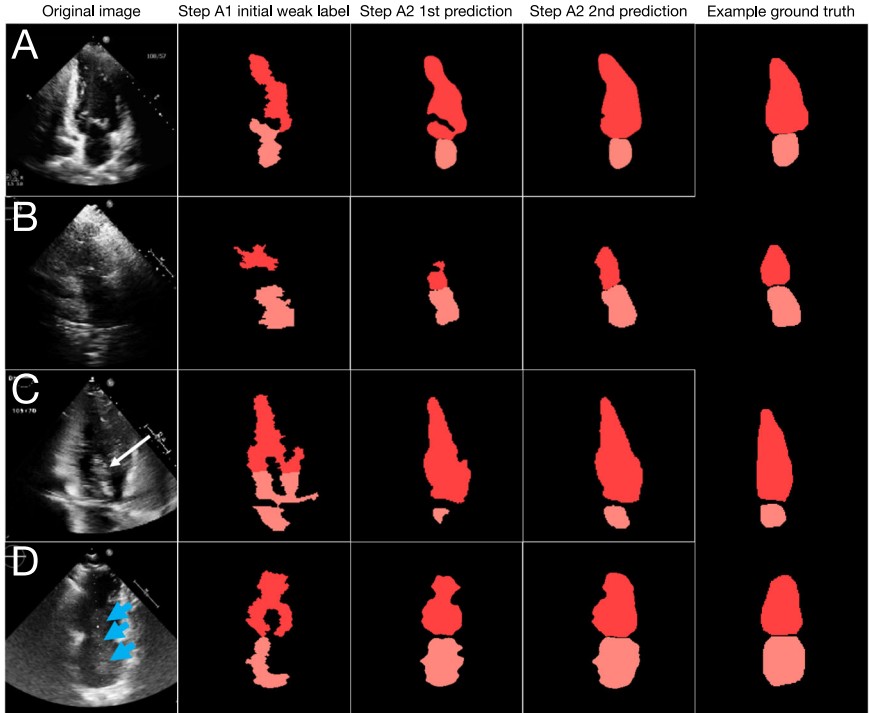

**Fig. 2 | Examples of segmentation and measurement improvement through successive steps of the SSL pipeline for the apical 2-chamber view.** Evolution of segmentation for several examples for A2C (**A**–**D**). **A** Shows a typical good-quality A2C image: the initial weak label (step A1) segments the chambers poorly but are corrected through successive steps of the pipeline to final prediction (step A2). The pipeline performs well even with left atrial enlargement in a technically difficult image (**B**), an image where LV contrast (white arrow) obscures the LV lumen (**C**), and an image with a large LA mass (blue arrowheads) prolapses into the LV (**D**). For each of (**A**–**D**), example human segmentations are shown as a visual aid for readers unfamiliar with echocardiograms; note that no manual segmentations were used in training the pipeline. Note too that clinical measurements were not made from a single image frame but rather all image frames in a given cardiac cycle, and often multiple views at once. A2C apical 2-chamber, LV left ventricle.

To demonstrate incremental impact of each step of the pipeline on prediction performance and generalizability, segmentations from intermediate steps of the pipeline were compared to clinical area measurements on images from the validation dataset (Fig. 5). The r² on chamber areas ranged from 0.06 to 0.22 when using initial weak labels compared to 0.53–0.81 using the full pipeline. Bias and LOA similarly improved with successive training steps.

## Pipeline-derived structure and function measurements are comparable to clinical echocardiogram measurements

Pipeline-derived measurements in the all-comers dataset were compared to clinical echocardiogram measurements (Fig. 6, Supplementary Table S1, Supplementary Figs. S3, S4).

**Left ventricle.** Pearson correlations (r) between the AI pipeline and clinical echocardiogram measurements for LV end-diastolic volume (LVEDV), LV end-systolic volume (LVESV), and LV ejection fraction (LVEF) were 0.84, 0.9, and 0.81, respectively (Supplementary Table S1). r² for these measurements were 0.70, 0.82, and 0.65; notably, the r² for LVEF achieved through the self-supervised AI pipeline is better than those reported from supervised learning[8,33] (Figs. 6A and 3C, Supplementary Table S1). Bland–Altman bias ± LOA (two standard deviations) for LVEDV, LVESV, and LVEF were 2.8 ± 51 mL, 5.3 ± 32 mL, and −5.3 ± 14.6%, respectively, similar to clinician variability studies[15,34–36] and consistent with those achieved from supervised learning[8,33] (Figs. 6A, 3C, Supplementary Table S1, Supplementary Fig. S3). Correlations for LV mass were lower (r = 0.74, r² = 0.55), but are still similar to reported benchmarks (Fig. 6D).

**Right ventricle.** r² were 0.69 and 0.71 for RVEDA and RVESA, respectively, indicating a strong correlation[37] between the SSL pipeline and clinical measurements. r's were 0.83 and 0.84 and bias ±LOA was −0.86 ± 5.4 cm², 1.6 ± 3.9 cm² (Supplementary Table S1, Figs. 6F, 3G, Supplementary Fig. S3).

**Atria.** The r² for left atrial volume was 0.84, showing a very strong correlation[37]. bias ±LOA was −0.15 ± 20 mL, consistent with clinical inter-observer bias and with supervised learning performance reported in the literature[8,34] (Fig. 3E). Right atrial volume r² was 0.76 and bias±LOA between SSL and clinical measurements was −2.1 ± 21 mL (Fig. 6H).

**Normal vs abnormal.** For further clinical context, the above measurements were indexed to body surface area where applicable and binned as normal or abnormal according to reference guidelines[3,38]. Accuracies and Cohen's kappa values are presented in Fig. 6, Supplementary Table S1 and Supplementary Fig. S4. Accuracy ranged from 0.71 for LV mass to 0.97 for LVEF. Kappa values ranged from 0.54 to 0.79 showing a moderate to substantial agreement[37] between AI pipeline and clinical echocardiogram measurements. LV mass and RVESA were the exceptions, where kappa values were only fair.

Taken together, measurements derived from self-supervised learning performed similarly to clinical inter- and intra-observer variability and to measurements derived from supervised deep learning.

## Self-supervised segmentation and clinical echo correlate similarly to CMR gold standard

Compared to all-comers, the CMR subset is younger, more male, and has more LV dysfunction (Table 1). Also, due to the nature of different clinical imaging protocols, CMR studies did not include measurements for all chambers. Furthermore, methods for measurement differ

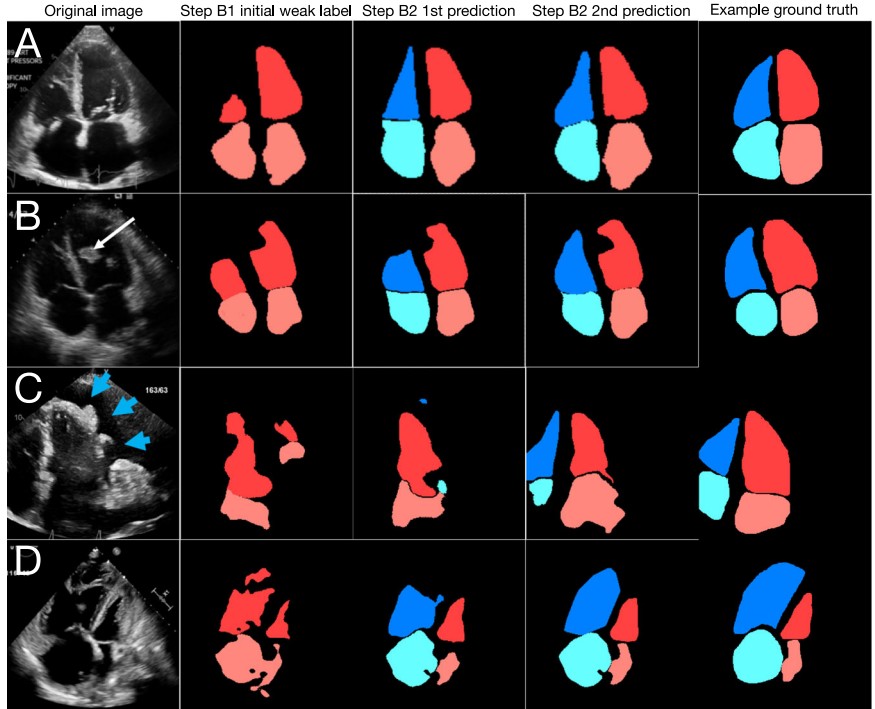

**Fig. 3 | Examples of segmentation and measurement improvement through successive steps of the SSL pipeline for the apical 4-chamber view.** Evolution of segmentation for several examples for A4C (**A–D**). **A** Shows a typical good-quality A4C image and its segmentation from the initial weak label (step B1) to final prediction (step B2). In (**B**), a filling defect from an LV thrombus (white arrow) was still present in the final segmentation. In (**C**), segmentation performs well despite presence of a pericardial effusion (blue arrowheads). In (**D**), segmentation performs despite RA and RV enlargement and septal flattening due to pulmonary hypertension. For each of (**A–D**), example human segmentations are shown as a visual aid for readers unfamiliar with echocardiograms; note that no manual segmentations were used in training the pipeline. Note too that clinical measurements were not made from a single image frame but rather all image frames in a given cardiac cycle, and often multiple views at once. A4C apical 4-chamber, LV left ventricle, RA right atrium, RV right ventricle.

between echo and CMR modalities. Despite this, correlations between SSL-derived measurements and CMR were similar to those of clinical echo measurements and CMR (Fig. 6, Supplementary Table S1).

Comparison of clinical echocardiogram measurements to CMR served as a benchmark. $r^2$ ranged from 0.67 to 0.77 for LV size and function. Bland–Altman bias±LOAs were −60 ± 96 mL, −34 ± 83 mL, 2.6 ± 20%, for LVEDV, LVESV, LVEF. These levels of agreement are similar to those reported in the literature[15,39], with echocardiography systematically underestimating LV systolic and diastolic size. LV mass showed poorer correlation with $r^2$ and Bias±LOA of 0.35 and 22 ± 130 g.

Comparing SSL-derived measurements to those from CMR showed similar to slightly worse correlations and similar limits of agreement (Fig. 6). $r^2$ for LV size and function ranged from 0.60 to 0.73. Bias ± LOA for LVEDV, LVESV, LVEF, and LV mass were −58 ± 109 mL, −30 ± 95 mL, −1.3 ± 22%, and 35 ± 122 g, respectively (Fig. 6D, Supplementary Table S1, Supplementary Fig. S3). As with the benchmark comparison of clinical echocardiography to CMR, LV mass from the SSL pipeline showed worse performance with $r^2 = 0.42$.

When indexed and binarized into normal vs. abnormal values, accuracies and kappas for LVEDV, LVESV, LVEF, and LV mass were the same or slightly better for the SSL pipeline than the clinical benchmark (Fig. 6A, 3D, Supplementary Fig. S4). Moreover, measurements derived from the SSL pipeline are more sensitive for abnormal biometrics (Supplementary Fig. S4).

### Performance across categorical thresholds and various study characteristics

While clinical guidelines do not report categorical thresholds for right heart measurements, left heart measurements can be further divided into normal, mild, moderate, and severe categories. We assessed categorical accuracy of LV measurements, comparing both clinical and SSL-derived measurments to the CMR gold standard where available (Supplementary Fig. S5A–E). As expected, accuracies for binary classifications were higher than categorical accuracies for both clinical and SSL-derived measurements ($p = 0.008$). However, there was no statistically significant difference between clinical and SSL-derived accuracies ($p = 0.66$). We further evaluated measurements by a range of study characteristics including age, gender, race, ethnicity, study quality, cardiac disease, and others (Supplementary Fig. S5F–U). As visualized by Bland–Altman analysis, performance did not differ significantly by these factors.

### AI pipeline performs well on external data

The external dataset offered the opportunity to calcuate Dice scores between SSL-predicted segmentation and manual tracings of the left ventricle, as well as comparison of LV function using estimations for LVEF (see Methods).

The average Dice score comparing SSL to manual tracing was 0.89 (95% CI [0.89]), representing good agreement. (For comparison, inter-observer variability in Dice scores on manual annotations of the left ventricle can range from 0.82 to 0.93[40].) Failures included examples where the external dataset was mislabeled, as well as correct external examples where the model failed (Supplementary Fig. S6).

With no A2C images available, we estimated LVEF using A4C images alone. When binarized by normal vs. abnormal, accuracy for LVEF was 0.79 compared to estimates provided for the external dataset. Notably, the SSL pipeline predicted segmentations for all four chambers, but labels were not available for LA, RV, or RA for performance evaluation.

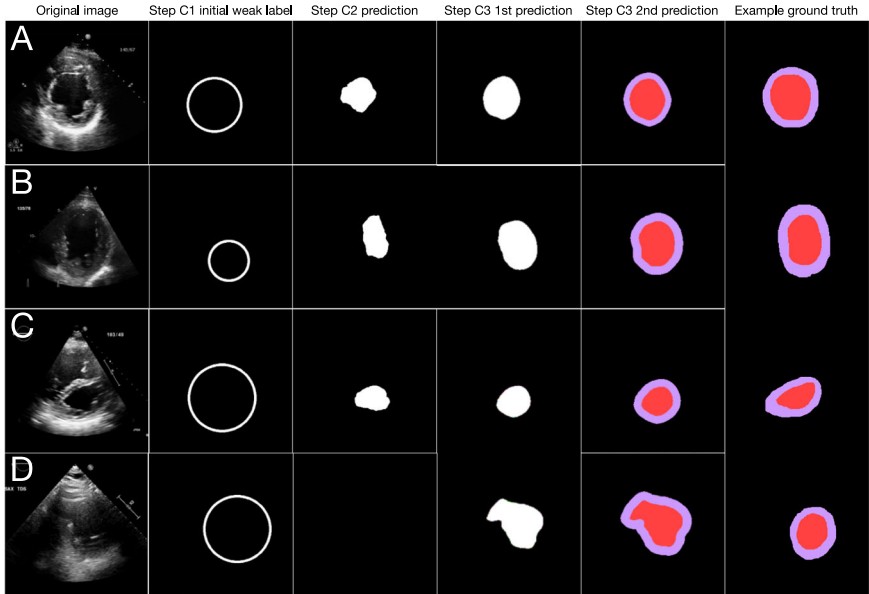

**Fig. 4 | Examples of segmentation and measurement improvement through successive steps of the SSL pipeline for the short-axis (SAX) view.** Evolution of segmentation for several examples for SAX view (**A**–**D**). **A** Shows a typical good-quality SAX image: the initial weak label (step C1) is a simple circle, but segmentations are built up through successive steps of the pipeline to final prediction (step C3). In (**B**), final segmentation prediction is reasonable despite a slightly off-axis SAX. In (**C**), segmentation performs despite pulmonary hypertension and septal flattening, but the true degree of septal flattening is blunted in the final prediction.

In (**D**), segmentation performs poorly due to a technically difficult image with dropout in the area of the septum but still recovers a final prediction despite no prediction in step C2. For each of (**A**–**D**), example human segmentations are shown as a visual aid for readers unfamiliar with echocardiograms; note that no manual segmentations were used in training the pipeline. Note too that clinical measurements were not made from a single image frame but rather all image frames in a given cardiac cycle, and often multiple views at once. SAX short axis.

## Discussion

Segmentation is a global, critical, and challenging task in ultrasound—exactly the sort of task that deep learning promises to help with. However, the noisy ultrasound modality presents a conundrum: it is recalcitrant to self-supervised learning, and yet supervised learning would require even more burdensome manual labeling. We solve this problem by developing a pipeline for self-supervised[1] segmentation of cardiac chambers from echocardiograms without any manual annotation or prompting. Self-supervised learning can facilitate the development medical foundation models without overburdening clinicians.

To demonstrate rigor and generalizability[31,41,42], we tested on large internal and external datasets—over 40 times the size of the training dataset. Furthermore, the all-comers dataset represented a full range of clinical characteristics and real-world image qualities—no image was excluded due to quality or pathology—making the fact that SSL pipeline's comparison to clinical measurements all the more impressive. Both internal and external datasets are enriched for cardiac diseases compared to the general population[43–46]. Several clinical measurements are a function of two (LA size) to four (LV function) successful segmentations, again making our pipeline's good agreement with clinical measurements noteworthy. Because the SSL pipeline uses information intrinsic to the image itself (whether the heart featured is normal or not), and because clinical information used was derived from all-comers rather than just normal cases, we see the SSL pipeline able to perform even when hearts may be significantly abnormal, such as those with CHD. We benchmarked our results against available measurements at several levels, including clinical echo measurements, CMR, inter-observer variability reported in the clinical literature, and supervised learning.

The scaling implications for self-supervised segmentation in echocardiography are clear. For our 450 training echocardiograms alone, we estimate (based on timed manual annotations of a small sample) that manually labeling all chambers in all three views would have taken a human 1664 h. The fact that both internal and external

datasets are missing many segmentations both systematically (e.g., right heart, left atrium) and randomly are testament to the laborious nature of segmentation in clinical practice as well. While manual segmentation scales poorly with each additional chamber, our human-label free pipeline is able to segment all chambers in an image simultaneously: for example, we predicted segments for all four chambers in the external dataset even though it only had left ventricular manual labels. Self-supervised segmentation in ultrasound has the potential to impute segmentations for large datasets, with beat-to-beat and even frame-to-frame granularity as previously demonstrated[47], for both clinical and research use.

At the same time, the efficiency and interpretability are greater than other current models[48,49]. Human learners require about $10^2$ echocardiography studies to become proficient[50]. Thus, our training dataset size of 450 studies approximates a human learner's efficiency, while our scalability (automatic measurements on >18,000 test studies) far outstrips human capability.

Where humans are inaccurate, variable, and incomplete in their measurements, this capability is an impactful step toward more complete, more reliable cardiac segmentation and measurement, which is a critical and clinically required component of every echocardiogram.

In this manuscript, we focused on measurements we could validate against available clinical measurements from our dataset, but a range of additional measurements are immediately possible. Scaling to segmentations of other anatomic structures, other views, and other types of ultrasounds are also feasible relatively quickly, using the same techniques demonstrated here.

In achieving self-supervised segmentation for echocardiograms, we demonstrate the effect of combining several deep learning techniques with traditional computer vision and with clinical spatial and geometric information. Computer vision preprocessing choices were driven by the physics of ultrasound imaging. We used two types of neural networks—segmentation networks that are known to detect textures, as well as edge-detection networks to strengthen

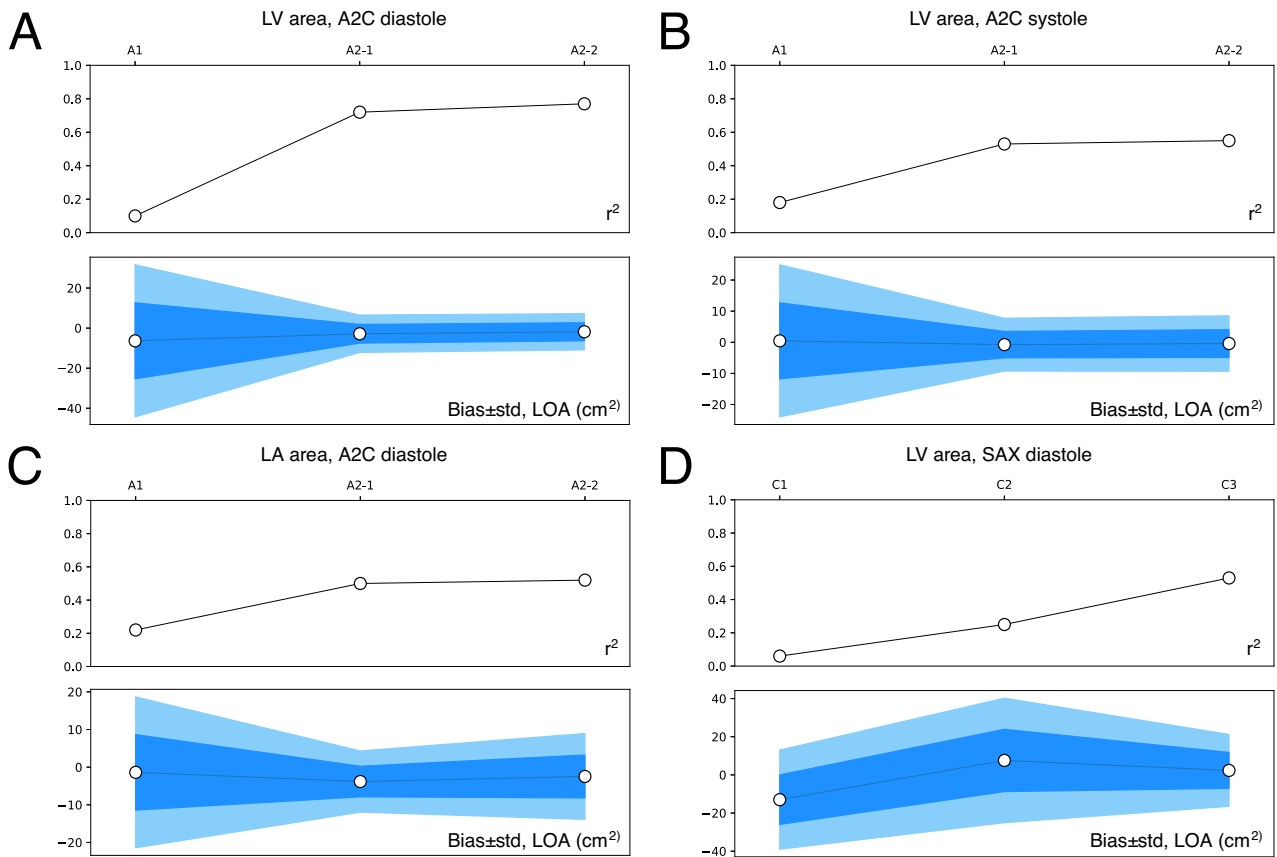

**Fig. 5 | Performance of segmentation and measurement through successive steps of the SSL pipeline. A–D** show area measurements calculated from segmentations at different steps in the pipeline, using images from the validation dataset. r² and Bland–Altman bias ± LOA comparing these measurements with clinical echocardiogram measurements improve across successive steps in the pipeline. Light blue, limits of agreement (two standard deviations); medium blue, one standard deviation. A2C apical 2-chamber, LV left ventricle, LA left atrium, SAX short axis, LOA limits of agreement. Source data are provided as a Source Data file.

ultrasound's noisy boundaries—to obtain better predictive performance than could be obtained from either one alone. Using a sequence of neural networks with leveraging early stopping and self-learning allowed each successive step in the pipeline benefitted from more data with cleaner labels. Even from a label as weak as a Hough circle for the SAX view, a reasonable LV segmentation was recovered, demonstrating the power of this approach. Finally, we found that using spatial modeling information in our pipeline as clinical domain knowledge improved performance[51].

Despite promising results, the SSL pipeline has also some limitations. While bias±LOA from the pipeline was comparable to clinical and supervised ML benchmarks, an ideal pipeline would have even tighter limits of agreement. From a practical standpoint, manual supervision of a strategically chosen[52] subset of images may improve on the results presented here; however, the purpose of this study was to demonstrate the potential of a completely human-label-free approach.

While not a function of the SSL pipeline itself, selection of image frames to serve as systolic and diastolic timepoints in real-world ultrasound clips is also a potential source of error (and an open problem which affects supervised segmentation as well[8]). If a clinician chose different systolic and diastolic frames for measurement, the final clinical measurement could differ from the AI pipeline even if frame-for-frame segmentations are highly concordant (as shown in the external dataset).

Another limitation not inherent to the SSL pipeline, but rather to the data, is that certain clinical echo measurements such as LV mass and right ventricle, are known to be more variable, contributing to lower observed performance for these measures. Repeated manual measurements for all chambers from multiple observers could better define the gold-standard echo measurement and reduce the potential effect of inter-observer measurement error on the clinical gold-standard, but doing this for thousands of test echocardiograms was not feasible within the scope of this study. Additionally, we look forward to further validation in primary care datasets and/or bespoke patient populations.

In summary, self-supervised segmentation of ultrasound represents a paradigm shift in how, rather than laboring to provide labels for data-hungry machine learning models, we can get machine learning to work for us efficiently, robustly, and scalably even for challenging imaging modalities, in order to solve important problems in cardiology and beyond.

## Methods
### Overview
Research was performed with waived consent in compliance with the UCSF IRB (retrospective records review). Echocardiogram images were used to develop a self-supervised pipeline (Fig. 1) for cardiac chamber segmentation of the apical-2chamber (A2C), apical 4-chamber (A4C), and short-axis mid (SAX) views. Initial weak labels derived from computer vision techniques together with aggregate statistical information about chamber shapes and relationships were created (Supplemental Methods). These were used to train the neural networks described below and in Fig. 1 in a series of early-learning and self-learning steps to arrive at a final prediction. The pipeline's final segmentation predictions were used to calculate structural and functional measurements according to clinical guidelines.

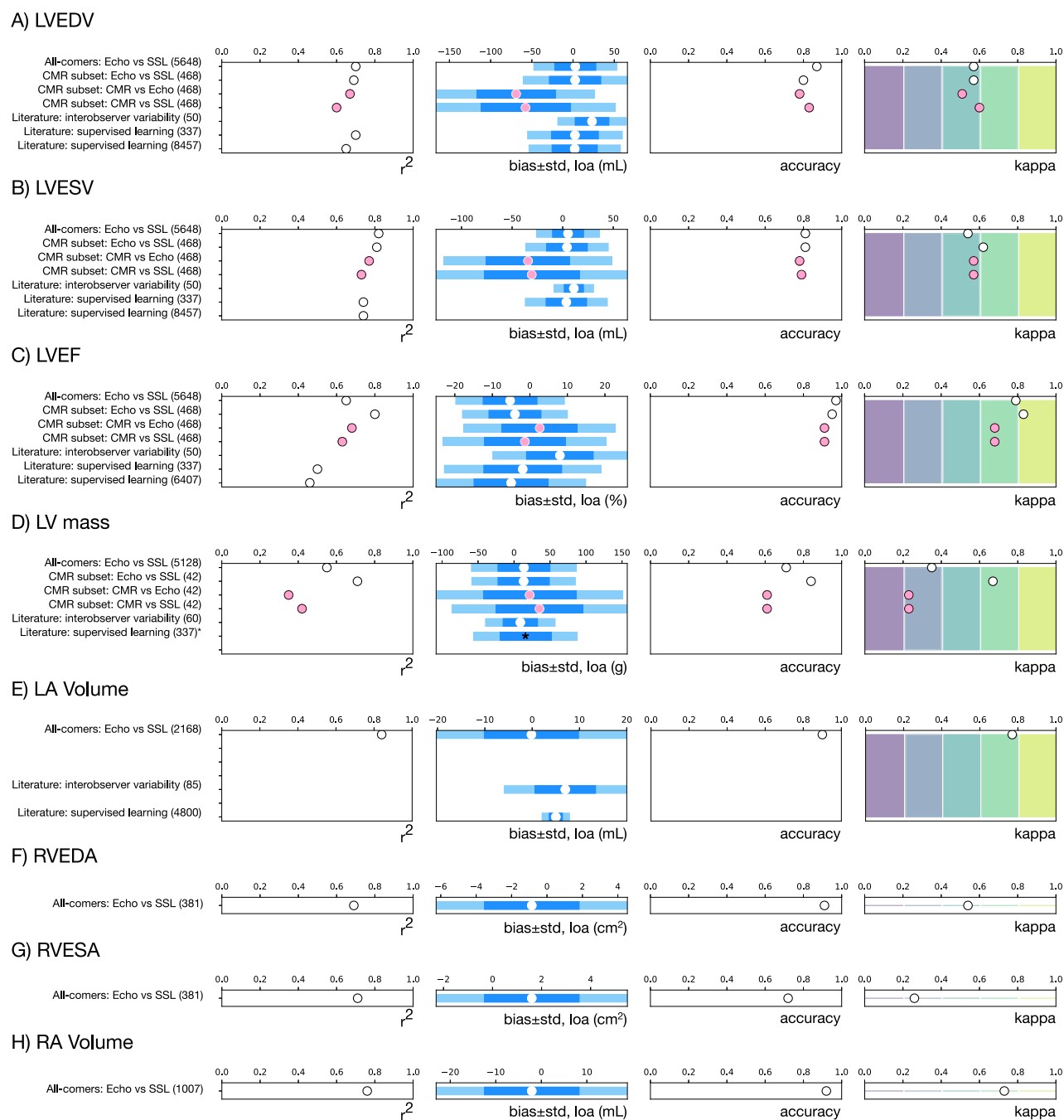

**Fig. 6 | Comparison of clinical and SSL-derived chamber measurements. A** LV end-diastolic volume (LVEDV), **B** LV end-systolic volume (LVESV), **C** LV ejection fraction (LVEF), **D** LV mass, **E** LA volume, **F** RV end-diastolic area (RVEDA), **G** RV end-systolic area (RVESA), and **H** RA volume. The datasets being compared are listed on the left, along with the number of datapoints in parentheses. Open (white) circles indicate intra-modality comparisons (echocardiography). Pink circles indicate comparisons across modality (to CMR). Light blue bars indicate Bland–Altman limits of agreement (LOA; two standard deviations); darker blue bars indicate one standard deviation. Black asterisk in (**D**) indicates an assumed Bland–Altman bias, as only LOA was reported. Vertical shading in the kappa comparison plots indicates ranges for poor (purple), fair (blue), moderate (teal), good (green), and excellent (yellow) agreement. Clinical intra-observer references include Jacobs et al. for LVEDV, LVESV, and LVEF[17]; Crowley et al. for LV mass[64]; and Mihaila et al. for LA volume. Supervised learning references include Ghorbani et al.[33] and Zhang et al.[8].

## Datasets

**Internal dataset.** A total of 8843 unique deidentified echocardiograms from UCSF were used (Table 1; self-reported gender was used). All clinically interpretable studies were included regardless of image quality. Top ten indications for study included arrhythmia/abnormal EKG (13%), heart failure/cardiomyopathy (13%), valve disease/murmur (11%), heart disease not otherwise specified (9%), CAD/chest pain (8%), chemotherapy monitoring (7%), perioperative assessment (6%), dyspnea (5%), hemodynamics (5%), and pulmonary hypertension/right heart failure (5%). A2C, A4C and SAX b-mode views were identified as previously described[53]. Views with color Doppler or LV contrast were excluded. For training and validation datasets, only images with a 200 mm field of view (FoV) were included. Only images from the first heart cycle were used to reduce image-level redundancy[52,54]. *Training and validation*. 2,228 videos (A2C, A4C, and SAX; 93,000 images) from 450 echocardiograms were used. Eighty percent were used for training

and 20% for validation. *Testing.* 8393 echocardiograms (4,476,266 images) were used as a holdout test set.

Measurements, such as left ventricle (LV) ejection fraction (LVEF), LV end-diastolic volume (LVEDV), LV end-systolic volume (LVESV), LV mass index (LVMI), left atrial (LA) volume, right atrial (RA) volume, and right ventricular end-diastolic area (RVEDA) were extracted from the echocardiogram database and used as ground-truth for performance evaluation for test echocardiograms. Where indicated, measurements were indexed by body surface area. Additionally, for echocardiograms with corresponding CMRs, corresponding measurements were extracted from the clinical CMR reports (thus, CMR-derived measurements were performed as per clinical guidelines for CMR). To evaluate test performance for a given chamber, only studies with available clinical measurements were used (Table 1, Supplementary Fig. S1). Training, validation, and test sets did not overlap by image, patient, or study.

**External dataset.** An external test dataset was obtained from (https://echonet.github.io/dynamic/). It consisted of 20,060 A4C images from 10,030 patients (one systolic and one diastolic image per patient), with manually annotated clinical tracings of the left ventricle, as well as a computer-estimated LVEF, LVESV, and LVEDV for each patient. Pixel size, alternative views, and manual tracings of LA, RA, and RV were not available.

## Data preprocessing and initial weak label extraction

The ultrasound region of interest was extracted from DICOM images and normalized to a size of 0.5 mm/pixel and resized to $256 \times 256$ pixels for segmentation networks or to $480 \times 480$ pixels for edge detection networks. Pixel intensities were normalized from 0 to 1. Computer vision and clinical knowledge techniques for initial weak label extraction are detailed in the Supplement. Preprocessing used Python 3.8 libraries OpenCV 4.2(opencv.org), scikit-image 0.16.2 (scikit-image.org), Scipy 1.4.1 (pypi.org/project/scipy/) and NumPy 1.16.2 (numpy.org).

## Neural network architectures

For each training step in Fig. 1, 80 percent of data was used for training and 20 percent for validation (split by patient). *Segmentation.* UNet is a neural network that has proved robust for segmentation in medical imaging. It was therefore used for segmentation as described[55] with the following modifications: the $1 \times 1$ output layer was sigmoid-activated, Adam optimizer was set with a learning rate of 1e−4, soft Dice loss and batch size 32 were used. Data augmentation consisted of random modifications to ~20 percent of training data as follows: rotations from 0 to 10 degrees, width and height shifts ±10% zoom ±20%, shear 0–0.03, horizontal flips, contrast stretching between 2nd and 98th percentile, cropping of 160×160 pixel patches, downscale from 0.25 to 0.5, pixel dropout of 0.1, median blur, Gaussian noise with zero mean and variance between 0.03 and 0.2, contrast limited adaptive histogram equalization with upper threshold value for contrast limiting of 0.02, and random tone curve with scale of 0.1.

**Edge-detection network.** In contrast to photographic images or drawings, objects in ultrasound images are known to have poor edges. Therefore, a holistically nested edge detection (HED) network was implemented for edge-detection tasks as described[56] with the following modifications: the model was initialized with ImageNet weights and fine-tuned using the same hyperparameters as in Xie et al.[57] with a batch size 8. The following data augmentations were randomly applied: rotations 0–10 degrees, width and height shifts ±10%, zoom ±8%, shear 0–0.03, and horizontal flips.

Models were implemented in Keras 2.3.0 (keras.io) with TensorFlow 2.1.4 (tensorflow.org) backend and trained in a NVIDIA Tesla M60 GPU with 8GiB of memory.

**Quality control (QC) for segmentations during network training.** At each step in the pipeline (Fig. 1), resultant segmentations were evaluated by shape descriptor analysis, discarding chambers of unreasonable size, eccentricity, or geometric chamber relationships as in Supplemental Methods.

**Early learning correction during network training.** Deep neural networks have been observed to be robust to label noise and first fit the training data with clean labels during the early learning phase, before eventually memorizing the examples with false labels or artifacts[58–62]. We therefore exploited early learning in training the segmentation networks used in this study. During network training, the transition between the early learning and memorization phases was detected by monitoring the soft Dice loss curve for the validation dataset. During network training, the soft Dice loss curve for the validation dataset was monitored in TensorBoard 2.3.0 (tensorflow.org). The transition from the transient phase (early learning) to the memorization phase was detected where the bend of the soft Dice loss curve in a shape of elbow is observed i.e., the point of maximal curvature using standard methods (e.g. kneed, pypi.org). Training was stopped at the elbow point (early stopping; number of epochs for each training step indicated in Fig. 1).

**Self-learning during network training.** In several neural network training steps (see Fig. 1), the neural network trained via early learning (with only a small number of examples that passed QC) was then used to infer on *all* available training and validation data in a self-learning manner[63], to recruit additional labeled training examples and higher-quality examples for a second round of training.

## Clinical calculations

The outputs of the segmentation pipeline were used to compute chamber dimensions, i.e., areas and volumes, for A2C, A4C and SAX views according to clinical guidelines[3] using the biplane method of discs for chamber volumes, and using the area-length method for LV mass, and the standard method for LVEF ([LVEDV-LVESV]/LVEDV) with the following exception: in the external dataset, LVEF was estimated from the A4C view only, as a ratio of pixel areas/pixel lengths.

For each metric, we inferred the segmentation model in all frames of all videos that included the chamber of interest. Systolic and diastolic frames were automatically detected by plotting areas of each chamber segmentation by frame for all videos[47] and a sinusoid with heart rate frequency was fitted to these data. Then, the $r^2$ between the area by frame and its sinusoid fitting was used to select the video with the best periodic function per patient. The frame with the largest area of the chosen video by the $r^2$ method was selected to be the diastole, and the smallest frame of the same video to be the systole. (This was done to avoid videos in which the heart may drift in and out of view, which is not uncommon in clinical practice.) This method failed (a sinusoid could not be fitted) on less than 2% of the entire dataset, in which cases diastole and systole frames were manually chosen.

## Statistical analysis

Pearson correlation was used to measure the linear relationship between chamber measurements. Linear regression analyses were performed to measure the strength of these relationships. Bland–Altman (BA) plots were analyzed to demonstrate the bias and 95% limits of agreement (two standard deviations) between different methods for each measurement. Cohen's Kappa was used to assess the agreement between the normal and abnormal values determined by different measurement methods (e.g., clinical vs SSL).

Mann–Whitney U testing was performed for patient demographics but not reported because due to dataset size, even minimal differences were found to be statistically significant. For r, $r^2$, and

Dice, 95% confidence intervals were computed by bootstrapping 10,000 iterations, however, CI were so narrow that they are not reported. All statistical testing was performed using the Scipy package (see above).

## Reporting summary

Further information on research design is available in the Nature Portfolio Reporting Summary linked to this article.

## Data availability

The dataset external dataset used to evaluate the findings of this study is publicly available at https://echonet.github.io/dynamic/ and can be used to reproduce all methods. Restrictions apply to the availability of the training dataset due to its personally identifiable nature and waived consent. We will consider requests to access the training data on an individual basis. Any data use will be restricted to non-commercial research purposes, and the data will only be made available upon execution of appropriate data use agreements. Source data are provided with this paper.

## Code availability

Code will be made available upon publication at github.com/ArnaoutLabUCSF/CardioML.

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

## Acknowledgements

We thank the patients, sonographers and clinicians whose work created the datasets used to develop and test this pipeline. We thank Yumiko Abe-Jones and Nader Najafi for help with data pulls. D.F. and R.A. were supported by the National Institutes of Health (R01HL150394) to R.A. R.A. is additionally supported by the Department of Defense (PR181763) and the Chan Zuckerberg Biohub.

## Author contributions

R.A. and D.F. conceived of the study. D.F. and C.L. performed data analysis with input from R.A. Z.S. performed data acquisition with input from R.A. R.A. and D.F. wrote the manuscript with input from C.L. and Z.S.

## Competing interests

The authors declare no competing interests.
