## [Transparent Peer Review file · Nature Communications]

Self-supervised learning for label-free segmentation in cardiac ultrasound

Corresponding Author: Dr Rima Arnaout

Version 1:

Reviewer comments:

Reviewer #1

(Remarks to the Author)

I was asked by the Editorial team to verify that the the authors had appropriately responded to the reviewers comments as the original reviewer have not responded. In addition I have also carefully gone over the manuscript.

My understanding is that albeit the authors have not done what the reviewer asked for. However, I do feel that the original reviewers wishes of including many more subjects is warranted. The main focus of the paper is in my understanding that it is possible to get clinically applicable segmentation methods based on a weak supervised approach. I do feel that the authors have done their best to clarify and demonstrate that the testing population is appropriate and covers a wide range of pathologies and diseases.

I feel that the proposed methodology is novel and interesting and warrants publication. As this is late in the review process and the paper is well written, I do not feel inclined to come with new suggestions that the authors must take into account.

However, I am curious about what the authors see regarding weak supervision, version an approach where the network is initially trained on a few number of manually outlined cases. This original network is then used in a bootstrap fashion to generate more training data where the user only need to make corrections and not generate the ground truth data from scratch. Maybe this could be discussed in the discussion. From my point of view I do not need to see another version of the paper.

REVIEWERS' COMMENTS

Reviewer #1 (Remarks to the Author): I was asked by the Editorial team to verify that the the authors had appropriately responded to the reviewers comments as the original reviewer have not responded. In addition I have also carefully gone over the manuscript.

>> Thank you for taking the time to review our manuscript.

My understanding is that albeit the authors have not done what the reviewer asked for. However, I do feel that the original reviewers wishes of including many more subjects is warranted. The main focus of the paper is in my understanding that it is possible to get clinically applicable segmentation methods based on a weak supervised approach. I do feel that the authors have done their best to clarify and demonstrate that the testing population is appropriate and covers a wide range of pathologies and diseases.

>> Thank you for recognizing the appropriateness and breadth of our extensive testing—the largest test set and highest test:train ratio in the field to date.

I feel that the proposed methodology is novel and interesting and warrants publication. As this is late in the review process and the paper is well written, I do not feel inclined to come with new suggestions that the authors must take into account.

>> Thank you for recognizing the novelty of our work and the quality of the manuscript.

However, I am curious about what the authors see regarding weak supervision, version an approach where the network is initially trained on a few number of manually outlined cases. This original network is then used in a bootstrap fashion to generate more training data where the user only need to make corrections and not generate the ground truth data from scratch. Maybe this could be discussed in the discussion. From my point of view I do not need to see another version of the paper.

>> We agree with the Reviewer, in fact we had the same thought for future work and we have mentioned this in the Discussion: “From a practical standpoint, manual supervision of a strategically chosen³² subset of images may improve on the results presented here; however, the purpose of this study was to demonstrate the potential of a completely human-label-free approach.” We have marked this in red for ease of viewing.